# Effects of Interface Induced Natural Strains on Magnetic Properties of FeRh

**DOI:** 10.3390/nano9040574

**Published:** 2019-04-09

**Authors:** Jeongmin Hong, Tiannan Yang, Alpha T. N’Diaye, Jeffrey Bokor, Long You

**Affiliations:** 1School of Optical and Electronic Information, Huazhong University of Science and Technology, Wuhan 430074, China; 2Department of Materials Science and Engineering, The Pennsylvania State University, University Park, PA 16801, USA; tuy123@psu.edu; 3Advanced Light Source, Lawrence Berkeley National Lab, Berkeley, CA 94720, USA; atndiaye@lbl.gov; 4Electrical Engineering and Computer Sciences, University of California Berkeley, Berkeley, CA 94720, USA; jbokor@eecs.berkeley.edu

**Keywords:** mixed states of FeRh, natural strains, strain modulation, FeRh, Phase Change, PC-memory

## Abstract

FeRh is a unique alloy which shows temperature dependent phase transition magnetic properties. The lattice parameter (*a*) of this CsCl-type (B2) structure is 4.1712 Å. It undergoes a first order transition from antiferromagnetic (AFM) to ferromagnetic (FM) order at around 370K and hysteretic behavior while cooling and heating. This meta-magnetic transition of FeRh is accompanied by an isotropic expansion in the unit cell volume, which indicates strong coupling between magnetic and structural properties of FeRh. Consequently, the magnetic and transport properties, such as magnetoresistance (MR), are changed during the transition. Due to its unique thermo-magnetic behaviors, FeRh is very important for future spintronic devices. The structure could be applicable for MR devices such as memory, sensors, and many other applications. It is critical to understand how to systematically influence phase transition of FeRh from naturally applied strains. Here, we investigate magnetic properties of FeRh in different strain environments induced by the substrates with different lattice parameters. The study was performed using synchrotron radiation, temperature dependent magnetometry, and magnetic scanning probe microscopy in addition to Landau theory calculations. We found that the naturally induced strains could modulate the magnetic phase locally and globally. The presence of the segments from the nucleation of the ferromagnetic domains, with a very thin layer in the antiferromagnetic matrix and the domain growth, were observed gradually. Using the systematic phenomena, it could be used for immediate applications in the future generation of phase change random access memory (PC-RAM) devices.

## 1. Introduction

FeRh is a thermo-magnetic alloy which changes its phase under the presence of strains due to the increase of the temperature. The transition has recently been shown to persist in thin film stacks and the substrate dependence of the transition has been exploited in hetero-structures of FeRh to induce the antiferromagnetic (AFM) to ferromagnetic (FM) transition [1,2,3,4,5,6,7]. The presence of the mixed phase could further develop a potential for controlling the transition through other effects. It is not clear about the mixed states, but transits the order completely by the strain.

Recently, electric field-induced phase transition in FeRh has been reported using piezoelectric substrates [8,9]. However, it is not systematically changed to FM states. It can be noted that there are two main reasons not to be controllable with the structures: (1) FeRh with the base substrates inducing strains are not controllable with the electric field; (2) FeRh thin films could introduce some other effects such as lattice mismatches and, arguably, others.

Epitaxial growth of FeRh onto different substrates could generate natural strains [9,10]. The phenomena show the origin of the transition systematically. Here, compressive strain results from different lattice structures. The substrate materials could induce the pre-defined strains naturally. To observe the systematic propagation after phase transition, we investigate mixed states which exist in both FM and AFM states together. The systematic strains would be important for the investigation of how strain induces the change of the magnetic properties.

The focus of this study here is to investigate the presence of the mixed states and find a way to control the states under the certain strain level. The separate effects of temperature and strain based on film thickness on the phase transition from AFM to FM of FeRh thin films by both experiment and Landau theory calculations were performed. Strain was introduced by epitaxial growth onto different substrates such as MgO, SrTiO_3_, La(Sr,Al)TiO_3_, LaAlO_3_, and YAlO_3_. The substrates’ thicknesses were increased from 20 to 50 nm. The observation was done with the natural strain-induced magnetization change by the different thickness and temperature.

During the hysteretic phase changing behaviors of FeRh, the structure could form the presence of the segments with north (N) and south (S) directions locally. Then, magnetic segments could propagate to the neighboring structures, which are a known mechanism of the phase changes [10]. However, the presence of mixed states, which are the co-existence of AFM and FM properties, shows even at high temperatures, 400K in this specific case. This study is to find the origin of the formation and propagation of the segments both through calculations and the experiment systematically using synchrotron radiation, temperature dependent magnetometry, and magnetic force microscopy studies [11,12].

The sensitivity is a consequence of the fact that the FM and AFM phases are relatively close in energy near the transition temperature, and thus can be inter-converted from one to the other by simply straining the lattice parameter of the system [13]. Artificially engineered heterostructures may enhance the control of resistance modulation, and therefore the structures using strain-engineered FeRh could be attractive to the immediate applications.

## 2. Results and Discussion

Naturally induced compressive strains result in lattice mismatch between FeRh and substrates. Eventually, it creates tensile strains, as shown in Figure 1a. The system designed has been built for four different types of structures, which represent different strains systematically. The structure exhibits natural strains by different substrates. Figure 1b shows naturally induced compressive strains are as follows: MgO, SrTiO_3_, La(Sr,Al)TiO_3_, LaAlO_3_, YAlO_3_, respectively. The strain levels of the materials show in the table. From the MgO to La(Sr,Al)TiO_3_ substrates, the strain is applied from 0.3% to 12% in a natural condition. LaAlO_3_ and YAlO_3_ exhibit rhombohedral and orthorhombic orientations, respectively. All others such as MgO, SrTiO_3_, and La(Sr,Al)TiO_3_ exhibit cubic crystal structure which is same as the structure of FeRh. We performed the thermodynamic stability analysis of the FM and AFM phases using the Landau theory of phase transitions. Free energies of FM and AFM phases are written as:(1)$$f_{FM}=\alpha_{0}\left(T-T_{C}\right)M^{2}+\alpha_{11}M^{4}+\frac{1}{2}c{\left(\varepsilon -\varepsilon^{0}\right)}^{2}-MH$$
(2)$$f_{AFM}=\beta_{0}\left(T-T_{N}\right)L^{2}+\beta_{11}L^{4}+\frac{1}{2}c{\left(\varepsilon -\varepsilon^{0}\right)}^{2}$$
where *M*, *L*, *ε*, and *ε*^0^ are magnetization field, AFM order parameter field, total strain, and transformation strain, respectively. Here we define $M=M_{Fe}+M_{Rh}$ and $L=M_{Fe}-M_{Rh}$, where *M_Fe_* and *M_Rh_* are the magnetization of the Fe sub-lattice and the Rh sub-lattice within the FeRh crystal, respectively. Through minimizing the total free energy of the system with respect to *M* and *L*, we obtain the equilibrium free energies of the FM and AFM phases under different substrate strains, respectively.

As shown in Figure 1c,d, under a given substrate strain the stable equilibrium state of the FeRh film manifests itself as either single phase AFM, single phase FM, or coexistence of AFM and FM phases (mixed states) caused by phase separation due to strain instability [14]. Figure 1e shows the calculated temperature–substrate strain phase diagram of FeRh films. At room temperature, upon an increasing tensile substrate strain, the FeRh film undergoes transitions from a pure AFM state to the mixed state, and then to a pure FM state.

Magnetometry measurements were performed to observe the temperature dependence. We checked four different temperature points (three points during heating and a point while cooling). The temperature dependent magnetometry clearly shows how varying temperature induces the magnetic property changes. Due to the small strain, MgO at room temperature shows ferromagnet behavior with small magnetization. When the temperature is increased, the increase of coercivity and magnetization was clearly observed. At the close of 400K, it also shows strong AFM and FM states, which is a strong indication of the mixed states. As shown in Figure 2, AFM signals at 400K are amplified. During the cooling process, the hysteretic behavior is observed. An FM state with higher magnetization has been shown. FeRh grown on MgO shows systematic transitions. The transition shows the increase of ***M_S_*** during heating, and in the case of cooling.

The presence of mixed states was observed above the bulk FM-AFM transition temperature (370K) as revealed by the calculations. The phenomena clearly indicate the lattice mismatch for the system. The temperature dependence was tested only for 10 nm FeRh on MgO substrates because natural strain due to lattice mismatch with temperature dependence could introduce multiple variables.

Magnetic force microscopy (MFM) characterize the magnetic properties of FeRh by varying the thickness. The MFM tip utilized had a magnetic coating layer of CoCr with an isolated island and the perpendicular magnetic anisotropy field to measure very low coercivity and magnetization. As shown in Figure 3a (top), for 20 nm of FeRh onto MgO, the magnetic phase exhibits that the segments appear. Also, Figure 3a (bottom) shows small signal of MFM phase is still observed in 50 nm of FeRh onto MgO. Meanwhile, the change of the roughness was observed as shown in Figure 3b due to the natural strains. In the case of SrTiO_3_ shown in Figure 3c (top and bottom), the MFM phases are similar, but slightly strong compared to the case of MgO. Even 50 nm of FeRh on STO still shows the segments and the surface is rougher than that of MgO case with 50 nm of FeRh.

For FeRh in the La(Sr,Al)TiO_3_ case, which shows the largest strain in cubic structure, it is not saturated even with the 50-nm-thick sample as shown in Figure 3e. A lot of segments form in 20 nm of FeRh on the bottom image of Figure 3e and propagate to form bigger domains of mixed states at 50 nm of FeRh onto LSAT. This is an indication of strain-induced magnetic properties change. As shown in Figure 3g–j, LaAlO_3_ and YAlO_3_ contain different crystal structures which show distinctive phenomena. The strain induces strong magnetization, but it stabilizes at 50 nm. For the YAO substrate, the largest segments were observed and small segments and AFM states still were observed at 50 nm range. From MFM measurements, saturation level of thickness is very different depending on the crystal structure and strain levels of the substrates. Segments show different magnetic characteristics which resulted from the natural strain. Surface roughness from topography measurements clearly shows systematic change from 20 nm to 50 nm, respectively.

X-ray Magnetic Circular Dichroism (XMCD) is the surface and element specific measurements which could probe top (less than 10 nm) of the structures at Fe *L*_3,2_ edges. The measurements are performed using total electron yield (TEY). Further to investigate the ferromagnetic properties of FeRh onto different substrates, we measured XMCD of the Fe *L*_3,2_-edges with different substrates. Figure 3k shows 20 nm of FeRh onto different substrates at 300K. For the cubic structure of the substrates, XMCD is not distinguishable, but absorption spectra show slight difference due to the lattice mismatch. However, rhombohedral structures such as YAlO_3_ and LaAlO_3_ intensify the XMCD signal, especially YAlO_3_ which has the highest strains applied. XMCD data exhibit different crystal structures and strains at room temperature.

The interface-induced strains-induced mixed magnetic states were investigated. We observed the effects by systematically growing the FeRh structures. The scenario, we observed, was as follows: the local strain resulted in magnetic segments which propagate further through the plane. However, once the thickness of the materials increased, the nature of FeRh came back to the normal AFM condition at room temperature. Magnetic force microscopy shows clear thickness dependence of the structures. Strains only made an effect on the thin layer of FeRh which immediately induced the transition and then it stabilized as weak FM/AFM states. The process starts from the initial presence of the segments, propagation of larger segments, and the merger of AFM domain with neighboring FM, which proves a clear mixed state.

## 4. Conclusions

In conclusion, we observed the origin of mixed states which are in the process of the propagation of the FM segments. The mixed states caused by lattice mismatches resulted in strains. This unique property of FeRh could be of immediate use for the future generation of multi–functional devices after engineering strains.

## Figures and Tables

**Figure 1 nanomaterials-09-00574-f001:**
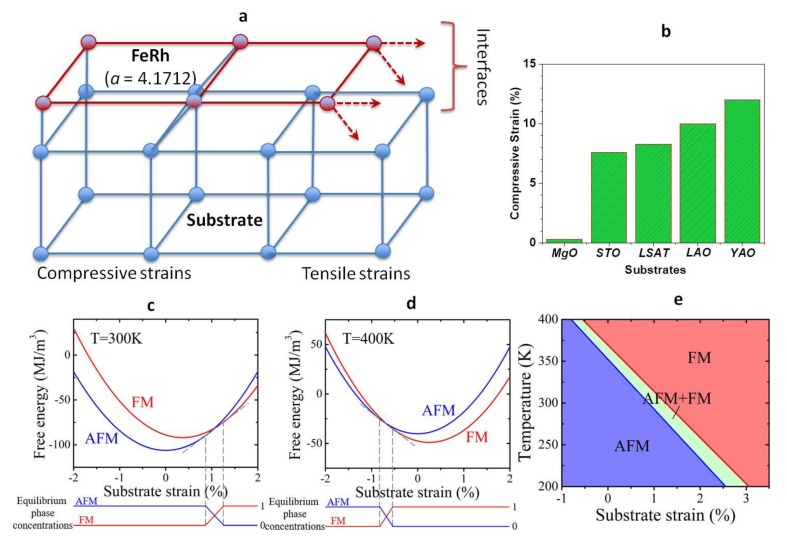
Model structures. (**a**) Schematics of the lattice deformation of the FeRh films. (**b**) Naturally induced strains of FeRh films on different substrates. (**c,d**) Calculated free energies of the ferromagnetic (FM) and antiferromagnetic (AFM) phases and their equilibrium phase concentrations as a function of substrate strains at (**c**) 300K and (**d**) 400K, respectively. (**e**) Calculated temperature–substrate strain phase diagram.

**Figure 2 nanomaterials-09-00574-f002:**
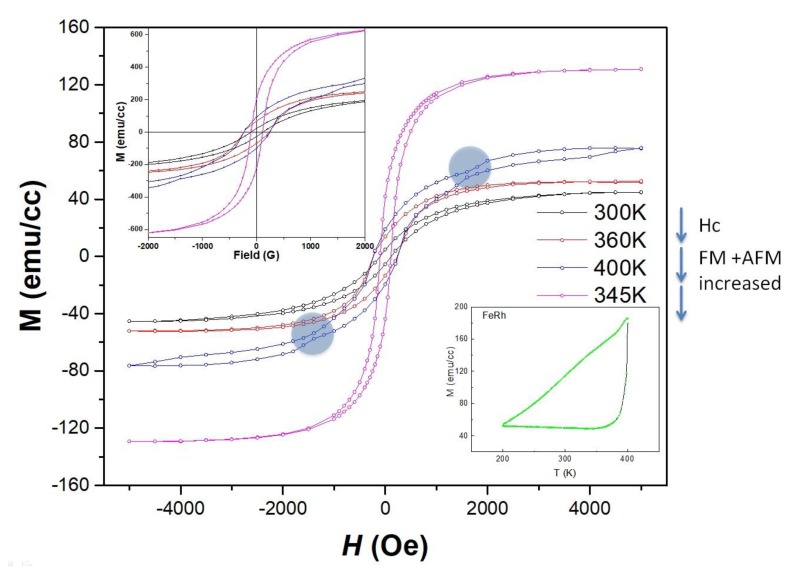
Temperature dependent m-H loop measurements. From room temperature, it was increased from room temperature (black) to 360K (red), 400K (blue) and then decreased down 345K (magenta).

**Figure 3 nanomaterials-09-00574-f003:**
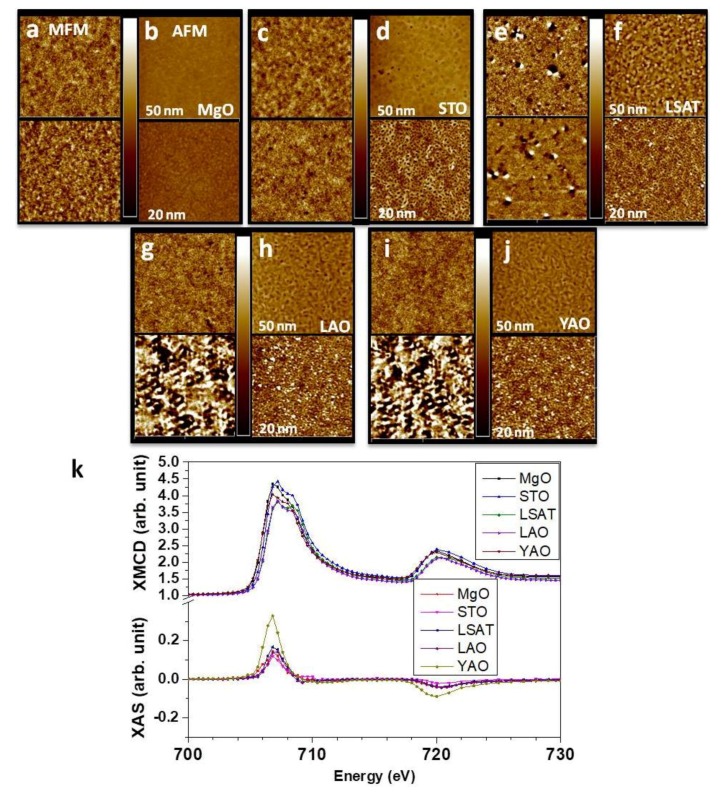
Magnetic properties of FeRh onto different substrates. (**a**) MFM measurements of MgO with two typical thickness, 20 nm (bottom) and 50 nm (top), respectively. (**b**) The corresponding AFM images of MFM of MgO. (**c**) MFM and (**d**) AFM of SrTiO_3_. (**e**) MFM and (**f**) AFM of La(Sr,Al)TiO_3_. (**g**) MFM and (**h**) AFM of LaAlO_3_. (**i**) MFM and (**j**) AFM of YAlO_3_ substrate. Top images were measured at 50 nm and bottom images at 20 nm thickness of the substrates. The size of the images is 5 × 5 µm^2^ (**k**) XMCD measurements with different substrates: MgO, SrTiO_3_, La(Sr,Al)TiO_3_, LaAlO_3_, and YAlO_3_, respectively.

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
