# Peer review of "Effects of Interface Induced Natural Strains on Magnetic Properties of FeRh"

_nanomaterials, 2019, doi:10.3390/nano9040574_

Round 1
Reviewer 1 Report
Comments:
Why did the Authors not perform tests for temperatures higher than 400 K? (Fig. 1e) Fig 2 (insert).
As the work indicates, measurements for different substrate thicknesses give different results. The paper shows the tests for substrate thickness of 20 and 50 nm, and the most important figure is made for substrate thickness of 10nm and only for one MgO compound.
Is it possible to perform the tests as in Figure 2 for the other substrates of different thickness.
The Authors state that the calculated FM-AFM transition temperature occurs at 370K (line 114). Why was not the experiment done for this temperature?
Why was not the AFM structure examined for a thickness between 20-50nm? In particular for Figures 3g and 3i.
Author Response
1. Why did the Authors not perform tests for temperatures higher than 400 K? (Fig. 1e) Fig 2 (insert).
è The main goal of this project is to observe the origin of the mixed states between antiferromagnetic and ferromagnetic behaviors. Above 400K, it is fully ferromagnet and so we decided no meaning above 400K.
2. As the work indicates, measurements for different substrate thicknesses give different results. The paper shows the tests for substrate thickness of 20 and 50 nm, and the most important figure is made for substrate thickness of 10nm and only for one MgO compound.
è We apologize to mislead the reviewer. One of the reasons to show about showing 20 nm and 50 nm of FeRh is that 10 nm thickness of the sample is very hard to control even with capping layer.
3. Is it possible to perform the tests as in Figure 2 for the other substrates of different thickness.
è Thank you for the comment and suggestion. Figure 2 shows the standard behavior of FeRh to the transition and others.
4. The Authors state that the calculated FM-AFM transition temperature occurs at 370K (line 114). Why was not the experiment done for this temperature?
è We apologize for not being very clear with this. The main point of the experiment is to find a way to the presence of mixed states from strains. The transition temperature at 370K from AFM to FM is being ignored because it shows temperature induced mixed states not strain induced states.
5. Why was not the AFM structure examined for a thickness between 20-50nm? In particular for Figures 3g and 3i.
è Thank you for pointing this out. The naturally induced strain dominantly shows FM states on those substrates.
Reviewer 2 Report
In this manuscript, the authors systematically investigate the influence of the strain by different substrates on the phase transition of FeRh. The authors found that the local strain induces the Ferro-magnetic domain and it disappears when the thickness of the materials increases. Their main results are supported by the measurement of the magnetic force microscopy which shows clear thickness dependence of the structures. However, I think the authors should deeply consider their results. They show AFM image of different film thickness. We can clearly find the change of the roughness due to the strains. It is well known that the disorder easily induces the phase transition, such as magnetic transition. So, we cannot distinguish that the effect on the Ferro-magnetic domain is caused by the strain or surface inhomogeneity. I think Discussion section is too short to compose a scientific paper. The authors should consider their results deeply, because their results are fine and systematic.
With above reason, I think this manuscript should be rejected from Nanomaterials.
Author Response
6. We can clearly find the change of the roughness due to the strains.
è Thank you so much for your suggestion. Yes, the change of the roughness from the strains is natural. However, we claim magnetic properties from the change in this manuscript.
7. It is well known that the disorder easily induces the phase transition, such as magnetic transition. So, we cannot distinguish that the effect on the Ferro-magnetic domain is caused by the strain or surface inhomogeneity.
è Thank you for pointing out the point. Yes, you are definitely true. It should be possible to have many factors. That is why we chose several different substrates with distinguishable lattice parameters. The thickness dependence shows systematic change due to the lattice mismatch.
8. I think Discussion section is too short to compose a scientific paper. The authors should consider their results deeply, because their results are fine and systematic.
è We added more description in discussion section.
Round 2
Reviewer 1 Report
Thank you for your comments
Reviewer 2 Report
Revised version of the manuscript has become fine. I understand the author’s claim. The authors measured with different substrates. The results are systematic and reliable. So, I think this manuscript should be accepted. But I just let you know that the roughness (disorder) would induce the phase transition. For example, in iron arsenide superconductor, AF order is enhanced by disorder (impurity) [Phys. Rev. Lett., 110, 207001 (2013)]. Situation is something different from this material, the same mechanism would be applied to this material.